# Effectiveness of a Standardized Nursing Process Using NANDA International, Nursing Interventions Classification and Nursing Outcome Classification Terminologies: A Systematic Review

**DOI:** 10.3390/healthcare11172449

**Published:** 2023-09-01

**Authors:** Claudio-Alberto Rodríguez-Suárez, Héctor González-de la Torre, María-Naira Hernández-De Luis, Domingo-Ángel Fernández-Gutiérrez, Carlos-Enrique Martínez-Alberto, Pedro-Ruymán Brito-Brito

**Affiliations:** 1Research Support Unit, Insular Maternal and Child University Hospital Complex, Canary Health Service, 35016 Las Palmas de Gran Canaria, Spain; 2Nursing Department, Faculty of Healthcare Science, Universidad de Las Palmas de Gran Canaria (ULPGC), 35016 Las Palmas de Gran Canaria, Spain; 3El Doctoral Primary Health Care Centre, Canary Health Service, 35110 Las Palmas de Gran Canaria, Spain; nairahernandez@celp.es; 4Primary Care Management of Tenerife, Canary Health Service, 38004 Santa Cruz de Tenerife, Spain; dfernand@ull.edu.es (D.-Á.F.-G.); pbritobr@ull.edu.es (P.-R.B.-B.); 5Faculty of Healthcare Science, Universidad de La Laguna (ULL), 38200 Santa Cruz de Tenerife, Spain; 6Nuestra Señora de Candelaria School of Nursing, 38010 Santa Cruz de Tenerife, Spain; carlosenrique.martinezalberto@gmail.com

**Keywords:** standardized nursing terminology, nursing process, nursing care, effectiveness, systematic review

## Abstract

The decision-making in clinical nursing, regarding diagnoses, interventions and outcomes, can be assessed using standardized language systems such as NANDA International, the Nursing Interventions Classification and the Nursing Outcome Classification; these taxonomies are the most commonly used by nurses in informatized clinical records. The purpose of this review is to synthesize the evidence on the effectiveness of the nursing process with standardized terminology using the NANDA International, the Nursing Interventions Classification and the Nursing Outcome Classification in care practice to assess the association between the presence of the related/risk factors and the clinical decision-making about nursing diagnosis, assessing the effectiveness of nursing interventions and health outcomes, and increasing people’s satisfaction. A systematic review was carried out in Medline and PreMedline (OvidSP), Embase (Embase-Elsevier), The Cochrane Library (Wiley), CINAHL (EbscoHOST), SCI-EXPANDED, SSCI and Scielo (WOS), LILACS (Health Virtual Library) and SCOPUS (SCOPUS-Elsevier) and included randomized clinical trials as well as quasi-experimental, cohort and case-control studies. Selection and critical appraisal were conducted by two independent reviewers. The certainty of the evidence was assessed with the Grading of Recommendations Assessment, Development and Evaluation Methodology. A total of 17 studies were included with variability in the level and certainty of evidence. According to the outcomes, 6 studies assessed diagnostic decision-making and 11 assessed improvements in individual health outcomes. No studies assessed improvements in intervention effectiveness or population satisfaction. There is a need to increase studies with rigorous methodologies that address clinical decision-making about nursing diagnoses using NANDA International and individuals’ health outcomes using the Nursing Interventions Classification and the Nursing Outcome Classification as well as implementing studies that assess the use of these terminologies for improvements in the effectiveness of nurses’ interventions and population satisfaction with the nursing process.

## 1. Introduction

The nursing process (NP) is the most common way used by nurses to provide and document the actions of nurses through a scientific method to identify, diagnose, intervene in and resolve health issues in the population within the scope of their disciplinary field. The complexity of the NP involves problem solving, reflective judgement and decision-making to achieve desired outcomes through five sequential steps: assessment, diagnosis, planning, implementation, and evaluation [1]. Its implementation demands cognitive, psychomotor and affective skills and capacities that underlie the clinical reasoning and care provided by nurses [2]. Each stage of the NP involves carrying out strategies to address the observed phenomenon, from the aspects concerned to the establishment of clinical judgment, including the gathering of information and recognition of health patterns, along with decision-making to determine the main and secondary interventions required for its resolution [3]. The nursing clinical decision-making regarding diagnoses, interventions and health outcomes of individuals can be assessed through the records made by nurses in information systems using standardized language systems (SLSs). Therefore, the phenomena and activities of nurses can be defined and described using SLSs through the retrieval of data from electronic records [4].

The use of such nursing terminologies in the scientific literature has been variable, with up to 72% of published studies using NANDA International (NANDA-I) [5] or its combination with Nursing Interventions Classification (NIC) [6] and Nursing Outcome Classification (NOC) [7] thus establishing itself as the most widely used system by nurses in the international context [8]. Through the review of the scientific literature, it is possible to assess the nurses’ use of NANDA-NIC-NOC (NNN) in clinical practice, as such records made in the patients’ clinical history provide evidence of the efficacy of the NP.

Two systematic reviews have recently been published that address the use of standardized nursing terminologies [9,10], but they have not focused on the exact topic of NNN terminologies. After a preliminary search of the scientific literature no other review has been found on the effectiveness of the NP using NNN in clinical practice. The only study that approaches this topic was conducted in 2017 by Sanson et al. [1] addressing a systematic review (SR) to understand the impact of nursing diagnoses on patient and organizational outcomes. These authors showed the existence of studies with methodological inconsistencies and an insufficient level of evidence (LE) about the impact of nursing diagnoses on patient and organizational outcomes [1].

For this assessment, the two following review questions were posed: does any association exist between the presence of related and risk factors and the clinical decision-making about nursing diagnoses? And, does the effectiveness of interventions, people’s health outcomes and people’s satisfaction increase when nurses use standardized NNN terminology? The research aims of this review are to synthesize the evidence on the effectiveness of the NP with standardized terminology using NNN in care practice to assess the association between the presence of the related and risk factors and the clinical decision-making about nursing diagnosis, and to assess the effectiveness of nursing interventions and health outcomes and increase people’s satisfaction.

## 2. Materials and Methods

An SR was carried out according to Joanna Briggs Institute (JBI) criteria; the reporting of results followed the Preferred Reporting Items for Systematic Reviews and Meta-Analyses (PRISMA), 2020 statement [11]. The research protocol was registered in the International Prospective Register of Systematic Reviews (PROSPERO); registration number CRD42020170350.

### 2.1. Sources of Information

The first step consisted of identifying previous publications on the subject of interest through various searches in PROSPERO and Google Scholar^®^ that could answer the research question. After this initial check, search strategies were employed in the following databases: Medline and PreMedline (through OvidSP), Embase (through Embase-Elsevier), The Cochrane Library (through Wiley), CINAHL (through EbscoHOST), SCI-EXPANDED, SSCI and Scielo (through WOS), LILACS (through the Health Virtual Library) and SCOPUS (SCOPUS-Elsevier). To complement these, manual searches were carried out in the Trip Database metasearch engine.

### 2.2. Search Methods

Searches were conducted on the 12 and 13 of January 2021 (Appendix A), establishing methodological limits to publications after 1992. Search strategies included the following terms: “nursing interventions classification” OR “nursing outcomes classification” OR “nanda international” OR “nnn terminology” in the title and abstract fields. Similarly, search strategies were adapted to each database. The search strategy was first checked by a documentalist in the Embase database (Appendix A) and independently reviewed by two of the authors. Once the definitive strategy was designed, it was adapted to the remaining databases selected.

### 2.3. Inclusion Criteria

Studies with the following design methodologies were included: Randomized clinical trials (RCT), quasi-experimental (non-randomized clinical trials and pre-post studies) and observational (cohort, case-control and case series), which consider the NP in English, Spanish and Portuguese language. Studies were included after 1992, coinciding with the year in which NNN terminology was officially recognized.

### 2.4. Exclusion Criteria

Other reviews (narrative reviews, scoping reviews, SR or umbrella reviews) and grey literature were excluded. Similarly, studies which did not consider the NP assessing the use of NNN were also excluded.

### 2.5. Quality Appraisal

The records were exported to an Excel^®^ spreadsheet for the selection process. Following the elimination of duplicates, studies were screened by title and abstract and classified into three groups: “potentially eligible”, “doubtful eligibility” and “excluded”. “Potentially eligible” and “doubtful eligibility” records were retrieved for full-text screening. The process was carried out by two independent reviewers and a third reviewer was consulted in the case of discrepancies. To determine study suitability, Critical Appraisal Skills Programme Español (CASPe) templates appropriate to each type of design were used so that for cohort studies, case-control studies and RCTs (11 items) scores ≤ 5 were considered low quality, scores 6–8 were considered moderate quality and scores ≥ 9 were considered high quality. To verify the suitability of the process, a pilot test was carried out on an initial record sample.

The certainty of the evidence (random sequence and allocation concealment), blinding bias of participants and researchers (concealment of allocation to study arm, intention to blind, method of blinding and blinding effectiveness), blinding bias to outcome assessors (reported, requiring researcher judgment or not requiring researcher judgment), attrition bias (incomplete data or omitted from analysis) and reporting bias (selective outcome reporting) were assessed, identifying each as: low risk, high risk, uncertain risk or not applicable. A pilot test of bias risk assessment was conducted on a sample of studies. Bias risk was considered in determining the degree of certainty of the evidence using the Grading of Recommendations Assessment, Development and Evaluation (GRADE) methodology.

### 2.6. Data Extraction

The research outcomes analysed, correspond to information on improvements in diagnostic association between the presence of the related and risk factors and the clinical decision-making about nursing diagnosis, effectiveness of interventions, health outcomes and people’s satisfaction. Separately, general study data were extracted. Data extraction was performed independently by two researchers and resolved through consensus with a third researcher in the case of discrepancies. The Mendeley^®^ bibliographic reference manager was used for data extraction and recorded in detail in the data extraction document. A pilot test of the extraction process was carried out on a sample of studies.

### 2.7. Data Synthesis

To organize the presentation of results, firstly, criteria established by JBI was followed to determine the LE for the effectiveness of each of the studies. The results were then organized according to the research outcomes below.

## 3. Results

The number of records identified was *n* = 4511; following elimination of *n* = 1601 duplicates, the number was *n* = 2910. During the title and abstract screening process *n* = 2820 were excluded, limiting the number of retrievable full-text records to *n* = 90. Of these, *n* = 4 could not be retrieved (1 was not retrieved due to conflict of references by the same title in 2 Digital Object Identifier (DOI) and different authorship names (Jones vs. Adams) in different journals; 3 were not retrieved due to impossibility to access the full text and no response after sending emails to the authors of correspondence) so that the number of studies assessed for eligibility was *n* = 86, of which *n* = 69 did not satisfy the inclusion criteria. Thus, the final number of included studies was *n* = 17, as can be seen in the flow chart below in Figure 1.

Following the screening process, those studies meeting the eligibility criteria were distributed among the authors for critical reading in pairs (CARS-CEMA; PRBB-MNHDL; and DAFG-HGDLT) and the subsequent measurement of interobserver agreement, through the determination of Cohen’s weighted kappa coefficient, are shown in Appendix A: Interobserver agreement on included studies. When the coefficient did not reach statistical significance, a third reviewer was consulted (CARS and MNHDL) to resolve agreement discrepancies.

All the studies showed high or moderate quality following critical reading with CASPe. The studies that showed high quality were the RCT (score 9/11) by Corcoles et al. [12], Guerra et al. [13], Gencbas et al. [14] and Sampaio et al. [15]. The remaining studies showed moderate quality in Appendix A: Critical reading scores for the included studies.

With regard to the design methodology, the studies included nine experimental designs (five RCT, one pseudo RCT and three quasi-experimental) and eight observational (one case control and seven cohort), which are shown together with sociodemographic characteristics in Table 1.

Following the GRADE methodology criteria, the overall quality of the certainty of scientific evidence was determined for each of the outcomes assessed. GRADE stipulates that studies with experimental designs show greater initial certainty, while observational studies do so with lesser initial certainty, although following application of compensation criteria for lowering or raising the quality of this initial certainty corresponding to each of the GRADE domains, their estimation is corrected. Final certainty was shown to be high in the study outcomes by Corcoles et al. [12], Silva et al. [16], Pascoal et al. [17], Silva et al. [18], Pascoal et al. [19], Reis and Jesus [20] and Pascoal et al. [21]. JBI criteria were simultaneously applied to assign the LE to each one, as shown in Appendix A: JBI level of evidence and degree of certainty using GRADE methodology.

Regarding research outcomes, the included studies assessed improvements in diagnostic accuracy (*n* = 6) and in people’s health outcomes (*n* = 11). No studies were identified that assessed outcomes in the efficacy of interventions or improvements in population satisfaction.

### 3.1. Diagnostic Etiological Association and Accuracy of Defining Characteristics

Studies assessing diagnostic indicators of NANDA-I determined the association with related/risk factors (RFs) (*n* = 3) and accuracy of defining characteristics (DCs) (*n* = 3).

The NANDA-I nursing diagnoses that addressed the etiological association of RFs were: risk of delayed surgical recovery (00246), dysfunctional ventilatory response to weaning (00034) and risk of falls (00155). The effect measures of these RFs were found to be statistically significant in most of the etiological indicators assessed, as shown in Table 2.

The articles that assessed the accuracy of the DCs (*n* = 3) concerned the NANDA-I nursing diagnoses: impaired gas exchange (00030), ineffective airway clearance (00031) and ineffective respiratory pattern (00032), as shown in Table 3.

### 3.2. People’s Health Outcomes

Articles that addressed effectiveness in people’s health outcomes did so from two perspectives.

First, regarding the general aspects of effectiveness (*n* = 2). On the one hand, with respect to the assessment of care planning using NNN and, on the other hand, concerning clinical reasoning. The study carried out by Cárdenas-Valladolid et al. [22] evaluated the implementation of care planning in primary care centres using standardized NNN terminology in the intervention group (IG) compared to the usual recording of non-standardized care as a control group (CG) through the prospective follow-up of a cohort (*n* = 23,488) over 2 years, demonstrating that both groups experienced a moderate reduction in cardiovascular risk factors observed at 12, 18 and 24 months for systolic blood pressure (SBP), diastolic blood pressure (DBP), glycosylated hemoglobin (HbA1c), LDL cholesterol and body mass index (BMI). The effect measure improved in the IG for all outcomes except LDL cholesterol and DBP. Following adjustment of the reference parameters for age, sex, type of treatment and physical activity, a reducing effect was observed in all outcomes except HbA1c, which was statistically significant for DBP (mean = −0.33 (CI = −0.63–0.04); *p* = 0.02). In general, the changes in the values for SBP, DBP, HbA1c, LDL cholesterol and BMI were greater in the IG than the CG, despite only reaching statistical significance in favour of the IG in HbA1c (*p* < 0.01), while the CG reached statistical significance in SBP (*p* < 0.01).

With regard to clinical reasoning, Müller-Staub et al. [23] developed a training program for nurses using guided clinical reasoning as an IG, compared with nurses who received training through classic discussion of clinical cases as a CG, showing greater acquisition of critical thinking skills for the application of NNN in clinical practice in the IG due to better internal consistency between diagnoses, interventions and outcomes, as shown in Table 4.

Secondly, studies that assessed the effectiveness of health outcomes in specific situations (*n* = 9) corresponded to the NANDA-I nursing diagnoses: functional urinary incontinence (00020), risk of falls (00155), ineffective health management (00078), risk of perioperative postural injury (00087), ineffective airway clearance (00031), nutritional imbalance: less than the body needs (00002), anxiety (00146) and sleep pattern disorder (00198). These studies assessed the interrelationship of NANDA-I diagnosis with respect to NIC and NOC terminologies. On the other hand, Guerra et al. [13] did not use NOC terminology to measure the effect of fall prevention on the reduction in risk of falls, while Bjorklund-Lima et al. [24] assessed the risk of perioperative postural injury using various NOCs but without reporting the NICs performed in the NP.

The statistically significant effect measures for each of the indicators of effectiveness on improving people’s health outcomes are shown in Table 5.

**Table 1 healthcare-11-02449-t001:** Sociodemographic characteristics of the included studies.

Author (Year)	Country	Methods	*n*	Study Period	Age
Corcoles et al. (2021) [12]	Spain	RCT	109	4 months	>65 years
Guerra et al. (2021) [13]	Brazil	RCT	118	10 months	>65 and <75 years
Lemos et al. (2020) [25]	Brazil	Quasi-experimental	28	9 months	Non-specific
Rembold et al. (2020) [26]	Brazil	Case control	239	6 years	>18 years
Silva et al. (2020) [16]	Brazil	Cohort	93	1 year	>18 years
Bjorklund-Lima et al. (2019) [24]	Brazil	Cohort	50	3 months	Non-specific
Pascoal et al. (2019) [17]	Brazil	Cohort	136	6–10 days	<5 years
Silva et al. (2019) [18]	Brazil	Quasi-experimental	101	1 year	>18 years
Vázquez-Sánchez et al. (2019) [27]	Spain	RCT	106	4 months	>18 years
Gencbas et al. (2018) [14]	Turkey	Pseudo RCT	62	Non-specific	Women (non-specific)
Sampaio et al. (2018) [15]	Portugal	RCT	74	6 months	>18 and <65 years
Pascoal et al. (2016) [19]	Brazil	Cohort	163	6–10 days	Children (non-specific)
Reis and Jesus (2015) [20]	Brazil	Cohort	271	5 months	Institutionalized elder patients (non-specific)
Pascoal et al.(2014) [21]	Brazil	Cohort	136	10 days	<5 years
Laguna-Parras et al. (2013) [28]	Spain	Quasi-experimental	291	14 months	>18 years
Cárdenas-Valladolid et al. (2012) [22]	Spain	Cohort	23,488	2 years	Non-specific
Müller-Staub et al. (2008) [23]	Switzerland	RCT	444	17 months	Non-specific

**Table 2 healthcare-11-02449-t002:** Statistically significant effect measures for the diagnostic etiological association with related/risk factors.

Author (Year)	Diagnostic Label	Related/Risk Factors	Effect Measures of Related/Risk Factors
Rembold et al. (2020) [26]	Risk of delayed surgical recovery (00246)	Pain	OR: 3.7 (CI: 2.04–6.65); *p* < 0.001
		Malnutrition	OR: 8 (CI: 1.96–32.60); *p* = 0.004
		Emotional responses recorded by nurses	OR: 5.2 (CI: 1.26–21.45); *p* = 0.020
		Impaired mobility	OR: 2.6 (CI: 1.42–4.71); *p* = 0.002
		Surgical wound infection	OR: 4.6 (CI: 2.03–10.47); *p* < 0.001
		Preoperative infection of surgical wound	OR: 7.6 (CI: 2.82–20.69); *p* < 0.001
		Prolonged surgical procedure	OR: 2.9 (CI: 1.61–5.20); *p* < 0.001
		Postoperative psychological disorders	OR: 6.4 (CI: 1.23–34.27); *p* = 0.023
		Extensive surgical procedure	OR: 1.8 (CI: 1.04–3.20); *p* = 0.036
		Interoperative complications	OR: 4.81 (CI: 1.55–14.92); *p* = 0.006
		Transfusion	OR: 4.25 (CI: 1.90–9.49); *p* < 0.001
		Anaemia	OR: 3.13 (CI: 1.65–5.93); *p* < 0.001
		Advanced cancer	OR: 2.87 (CI: 1.06–7.77); *p* = 0.032
Silva et al. (2020) [16]	Dysfunctional ventilatory response to weaning (00034)	Water balance	(Pre) M: 1.64; SD: 13.04.
			(Post) M: 13.04 SD: 13.14
			OR: 1.08 (CI: 1.03–1.12); *p* = 0.000
		Quantity of antibiotics administered	(Pre) M: 1.02; SD: 1.00
			(Post) M: 2.20; SD: 1.17
			OR: 2.56 (CI not reported); *p* = 0.000
		Age	(Pre) M: 56.85; SD: 18.48
			(Post) M: 65.76; SD: 18.53
			OR: 1.03 (CI: 1.00–1.05); *p* = 0.027
		Edema MI	(Pre) M: 1.02; SD: 0.94
			(Post) M: 2.39; SD: 1.56
			OR: 2.21 (CI: 1.53–3.19); *p* = 0.000
		Edema MS	(Pre) M: 1.23; SD: 1.02
			(Post) M: 2.34; SD: 1.56
			OR: 1.89 (CI: 1.34–2.66); *p* = 0.000
		Heart rate	(Pre) M: 85.73; SD: 18.07
			(Post) M: 96.42 SD: 16.40
			OR: 1.04 (CI: 1.01–1.06); *p* = 0.007
		Hemodialysis	(Pre) *n* = 8 (28.6%)
			(Post) *n* = 20 (71.4%)
			OR: 5.24 (CI: 1.98–13.83); *p* = 0.000
		Hyperthermia	(Pre) *n* = 5 (22.7%)
			(Post) *n* = 17 (77.3%)
			OR: 6.66 (CI: 2.19–20.24); *p* = 0.000
		Oliguria	(Pre) *n* = 5 (16.1%)
			(Post) *n* = 26 (83.9%)
			OR: 16.29 (CI: 5.32–49.93); *p* = 0.000
		Clinical severity on admission to ICU (SAPS 3)	(Pre) M: 54.52; SD: 13.13
			(Post) M: 64.39; SD: 17.06
			OR: 1.04 (CI: 1.01–1.08); *p* = 0.004
		Use of NIV (non-invasive ventilation) after extubation	(Pre) *n* = 10 (32.3%)
			(Post) *n* = 21 (67.7%)
			OR: 4.41 (CI: 1.75–11.09); *p* = 0.002
Reis and Jesus (2015) [20]	Risk of falls (00155)	History of falls	(Fall) *n* = 59 (85.51%)
			(No fall) *n* = 145 (71.78%)
			OR: 2.32 (CI: 1.11–4.85); *p* = 0.025
		Foot problems	(Fall) *n* = 26 (37.68%)
			(No fall) *n* = 40 (19.8%)
			OR: 2.45 (CI: 1.35–4.44); *p* = 0.003
		Polypathology	(Fall) *n* = 19 (25.54%)
			(No fall) *n* = 24 (11.88%)
			OR: 2.82 (CI: 1.43–5.56); *p* = 0.002
		Wandering	(Fall) *n* = 46 (66.67%)
			(No fall) *n* = 100 (49.5%)
			OR: 2.04 (CI: 1.15–3.61); *p* = 0.014
		Cerebrovascular accident (CVA)	(Fall) *n* = 25 (36.23%)
			(No fall) *n* = 48 (23.76%)
			OR: 1.82 (CI: 1.01–3.28); *p* = 0.045

FE: fixed effects; RE: random effects; OR: odds ratio; CI: confidence interval; M: mean; and SD: standard deviation.

**Table 3 healthcare-11-02449-t003:** Statistically significant effect measures for diagnostic accuracy of defining characteristics.

Author (Year)	Diagnostic Label	Defining Characteristics	Effect Measures of the Defining Characteristics
Pascoal et al. (2019) [17]	Impaired gas exchange (00030)	Abnormal skin color	RR: 1.54 (CI: 1.08–2.20); *p* = 0.016
		Hypoxemia	RR: 135.7 (CI: 75.10–245.19); *p* < 0.001
Pascoal et al. (2016) [19]	Ineffective airway clearance (00031)	Change in respiratory rate	OR: 2.88 (CI: 1.34–6.19); *p* = 0.007
		Cyanosis	OR: 0.03 (CI: 0.006–0.19); *p* < 0.001
		Difficulty vocalizing	OR: 10.04 (CI: 2.38–42.35); *p* = 0.002
		Open eyes	OR: 68.73 (CI: 1.53–3086.70); *p* < 0.001
		Adventitious lung sounds	OR: 300.58 (CI: 43.67–2068.86); *p* < 0.001
		Reduced breathing sounds	OR: 9.008 (CI: 2.75–29.48); *p* < 0.001
		Ineffective cough	OR: 129.53 (CI: 33.40–502.19); *p* < 0.001
Pascoal et al. (2014) [21]	Ineffective respiratory pattern (00032)	Altered respiratory depth	OR: 73.32 (CI: 15.45–347.79); *p* < 0.001
		Anteroposterior diameter increase	OR: 31.56 (CI: 7.20–138.34); *p* < 0.001
		Altered chest movements	OR: 259.14 (CI: 31.41–2137.92); *p* < 0.001
		Orthopnea	OR: 30.14 (CI: 4.49–202.43); *p* < 0.001
		Tachypnea	OR: 5.89 (CI: 2.02–17.11); *p* = 0.001
		Use of accessory muscles for breathing	OR: 2595.06 (CI: 343.88–19,583.3); *p* < 0.001

RR: relative risk; CI: confidence interval; and OR: odds ratio.

**Table 4 healthcare-11-02449-t004:** Statistically significant effect measures for overall effectiveness in health outcomes.

Author (year)	General Aspect Assessed	Indicator of Effectiveness	Effect Measures
Cárdenas-Valladolid et al. (2012) [22]	Care planning using NNN	Reduction in DAT in the IG	N: 4354 (Initial) M:76; SD: 10 (Final) M: 75; SD: 9 (Difference) M: −1.45; SD: 11 (AE) M: −0.33; IC: −0.63–0.04; *p* = 0.02
		Reduction in HbA1c (<7%) in the IG	Initial: 47.6% 24 months: 55.2% Change: 7.6% *p* < 0.01
		Reduction in SAT (<130 mmhg) in the CG	Initial: 31.6% 24 months: 35.5% Change: 3.9% *p* < 0.01
Müller-Staub et al. (2008) [23]	Nurses’ clinical reasoning	NANDA-I	Pre (IG) M: 2.69; SD: 0.9 Post (IG) M: 3.7; SD: 0.54 *p* < 0.0001
			Pre (CG) M: 3.13; SD: 0.89 Post (CG) M: 2.97: SD: 0.8 *p* = 0.17
		NIC	Pre (IG) M: 2.33: SD: 0.93 Post (IG) M: 3.88; SD: 0.35 *p* < 0.0001
			Pre (CG) M: 2.7; SD: 0.88 Post (CG) M: 2.46; SD: 0.95 *p* = 0.05
		NOC	Pre (IG) M: 1.53; SD: 1.08 Post (IG) M: 3.77; SD: 0.53 *p* < 0.0001
			Pre (CG) M: 2.02; SD: 1.27 Post (CG) M: 1.94; SD: 1.06 *p* = 0.62

NNN: NANDA-NIC-NOC; DAT: diastolic arterial tension; IG: intervention group; M: mean; SD: standard deviation; AE: adjusted effect; HbA1c: glycosylated hemoglobin; CG: control group SAT: systolic arterial tension; NANDA-I: NANDA International; NIC: Nursing Interventions Classification; NOC: Nursing Outcome Classification.

**Table 5 healthcare-11-02449-t005:** Statistically significant effect measures for people’s health outcomes.

Author (Year)	NNN Interrelationship	Indicator of Effectiveness	Effect Measure
Corcoles et al. (2021) [12]	NANDA-I Functional urinary incontinence (00020) NIC Urinary habit training (0600) NOC Urinary continence (0502)	3 months: Continence	No: 25.5% (IG) and 47.2% (CG) Yes: 74.5% (IG) and 52.8% (CG) RR = 0.54 (CI: 0.31–0.94); *p* = 0.022; NNT: 5
		3 months: Diurnal incontinence episodes	(CG) M: 1.54; SD: 2.26 (IG) M: 0.31; SD: 0.76 *p* = 0.002
		3 months: Nocturnal incontinence episodes	(CG) M: 0.79; SD: 1.29 (IG) M: 0.21; SD: 0.5 *p* = 0.012
		6 months: Continence	No: 25.5% (IG) and 49% (CG) Yes: 74.5% (IG) and 51% (CG) RR = 0.52 (CI: 0.3–0.9); *p* = 0.014; NNT: 4
		6 months: Diurnal incontinence episodes	(CG) M: 1.8; SD: 2.51 (IG) M: 0.54; SD: 1.46 *p* = 0.007
		6 months: Nocturnal incontinence episodes	(CG) M: 0.9; SD: 1.47 (IG) M: 0.35; SD: 0.86 *p* = 0.016
Guerra et al. (2021) [13]	NANDA-I Risk of falls (00155) NIC Fall prevention (6490)	Decreased incidence of falls	13.6% reduction in both groups (IG) 6.9% versus (CG) 20.0%; *p* = 0.038 34.48% reduction in relative risk of falls in the IG
		Cause of fall: difficulty walking	(IG) 0.0% versus (CG) 10.0%; *p* = 0.013
		Place where fall occurred: living room	(IG) 0.0% versus (CG) 13.3%; *p* = 0.004
Lemos et al. (2020) [25]	NANDA-I Ineffective health management (00078) NIC Teaching: disease process (5602) Teaching: prescribed medication (5616) Teaching: prescribed diet (5614) NOC Knowledge: heart failure management (1835) Knowledge: diabetes management (1820)	Knowledge: heart failure management	(1st assessment) M: 2.05; SD: 0.28 (2nd assessment) M: 2.54; SD: 0.30 (Difference) M: 0.48; SD: 0.21 *p* = 0.002
		Knowledge: diabetes management	(1st assessment) M: 2.61; SD: 0.55 (2nd assessment) M: 3.21; SD: 0.57 (Difference) M: 0.59; SD: 0.20 *p* = 0.000
Bjorklund-Lima et al. (2019) [24]	NANDA-I Risk of perioperative postural injury (00087) NOC Consequences of immobility: physiological (0204) Tissue perfusion: cellular (0416) Tissue perfusion: peripheral (0407) Thermoregulation (0800) Neurological status: peripheral (0917) Tissue integrity: skin mucous membranes (1101)	Measurement at five timepoints: mean scores in most NOCs decreased at timepoint 2 (T2-assessment in the operating room at the end of surgery) compared with timepoint 1 (T1-preoperative)	Most NOC showed improvement (*p* < 0.001) in postoperative time score (T3, T4 and T5) compared with T2
		NOC Consequences of immobility: physiological (0204)	T1 (M: 5.0; SD: 0.0), T2 (M: 4.0; SD: 0.0), T3 (M: 4.24; SD: 0.06), T4 (M: 4.80; SD: 0.05), T5 (M: 4.86; SD: 0.04); *p* < 0.001
		NOC Severity of blood loss (0413)	T1 (M: 4.59; SD: 0.04), T2 (M: 4.59; SD: 0.07), T3 (M: 4.58; SD: 0.09), T4 (M: 4.32 (SD: 4.32; SD: 0.10) T5 (M: 4.45; SD: 0.08); *p* = 0.014
		NOC Circulatory status (0401)	T1 (M: 4.59: SD: 0.06), T2 (M: 4.68; SD: 0.04), T3 (M: 4.41; SD: 0.07), T4 (M: 4.65; SD: 0.06), T5 (M: 4.43; SD: 0.08); *p* = 0.002
		NOC Tissue perfusion: cellular (0416)	T1 (M: 4.94; SD: 0.02), T2 (M: 4.68; SD: 0.05), T3 (M: 4.67; SD: 0.05), T4 (M: 4.68; SD: 0.04), T5 (M: 4.70; SD: 0.04); *p* < 0.001
		NOC Tissue perfusion: peripheral (0407)	T1 (M: 4.92; SD: 0.03), T2 (M: 4.31; SD: 0.09), T3 (M: 4.42; SD: 0.08), T4 (M: 4.58; SD: 0.06), T5 (M: 4.58; SD: 0.08); *p* < 0.001
		NOC Thermoregulation (0800)	T1 (M: 4.69; SD: 0.05), T2 (M: 4.69; SD: 0.05), T3 (M: 4.45; SD: 0.08), T4 (M: 4.86; SD: 0.03), T5 (M: 4.73; SD: 0.05); *p* < 0.001
		NOC Neurological status: peripheral (0917)	T1 (M: 4.96; SD: 0.03), T2 (M: 3.98; SD: 0.18), T3 (M: 4.39; SD: 0.15), T4 (M: 4.65; SD: 0.12), T5 (M: 4.76; SD: 0.11); *p* < 0.001
		NOC Tissue integrity: skin and mucous membranes (1101)	T1 (M: 4.93; SD: 0.02), T2 (M: 4.30; SD: 0.05), T3 (M: 4.50; SD: 0.05), T4 (M: 4.69; SD: 0.04), T5 (M: 4.71; SD: 0.04); *p* < 0.001
Silva et al. (2019) [18]	NANDA-I Ineffective airway clearance (00031) NIC Cough enhancement (3250) Ventilation assistance (3390) Airway management (3140) NOC Respiratory status (0415)	NIC Cough enhancement (3250): Respiratory rate	PR = 0.39 (CI: 0.81–0.98); *p* = 0.005
		NIC Cough enhancement (3250): Adventitious respiratory sounds	PR = 2.20 (CI: 2.55–8.11); *p* = 0.021
		NIC Cough enhancement (3250): Thoracic surgery patients: improvement in ability to eliminate secretions	PR = 4.55 (CI: 1.13–20.87); *p* = 0.0001
		NIC Cough enhancement (3250): Thoracic surgery patients: increase in ability to cough	PR = 4.75 (CI: 2.55–8.11); *p* = 0.024
		NIC Cough enhancement (3250): Abdominal surgery patients: reduction in the presence of dyspnea in mild exertion	PR = 0.38 (CI: 0.62–0.90); *p* = 0.022
		NIC Cough enhancement (3250): Abdominal surgery patients: decrease in changes in respiratory rate	PR = 0.25 (CI: 0.10–0.60); *p* = 0.001
		NIC Cough enhancement (3250): Abdominal surgery patients: decrease in nasal flaring	PR = 0.06 (CI: 0.006–0.74); *p* = 0.040
		NIC Cough enhancement (3250): Abdominal surgery patients: decrease in inspiration depth	PR = 0.45 (CI: 0.21–0.92); *p* = 0.028
		NIC Cough enhancement (3250): Abdominal surgery patients: improvement in adventitious respiratory sounds	PR = 2.82 (CI: 1.06–7.49); *p* = 0.031
		NIC Ventilation support (3390): Improvement in ability to eliminate secretions	PR = 0.14 (CI: 0.35–0.58); *p* = 0.009
		NIC Ventilation support (3390): Improvement in respiratory rate	PR = 0.43 (CI: 0.19–0.95); *p* = 0.034
		Ventilation support (3390): Improvement in inspiration depth	PR = 0.44 (CI: 0.20–0.97); *p* = 0.040
		NIC Ventilation support (3390): Abdominal surgery patients: decrease in use of accessory muscles	PR = 0.41 (CI: 0.16–1.007); *p* = 0.046
		NIC Airway management (3140): Decrease in accumulation of sputum	PR = 0.15 (CI: 0.30–0.76); *p* = 0.036
		NIC Airway management (3140): Improvement in adventitious respiratory sounds	PR = 0.14 (CI: 0.24–0.90); *p* = 0.047
Vázquez-Sánchez et al. (2019) [27]	NANDA-I Nutritional imbalance: lower than body needs (00002) NIC Nutritional assessment (5246) NOC Knowledge: Prescribed diet (1802) Indicator 180201: Prescribed diet Compliance behavior: prescribed diet (1622) Indicator 162202: Select foods and liquids compatible with prescribed diet	NIC increased NOC indicator score: prescribed diet	IG: 1.57 vs. CG: 0.22; *p* < 0.001
		NOC indicator: prescribed diet	Correlated with BMI (r = −0.34; *p* = 0.001), with Barthel index score (r = 0.50; *p* < 0.001) and with MUST questionnaire score (r = 0.28; *p* = 0.007)
		Intervention increased NOC indicator score NOC: select foods and liquids compatible with prescribed diet.	IG: 1.20 vs. CG: 0.26; *p* < 0.001
		NOC indicator: select foods and liquids compatible with prescribed diet	Correlated with BMI score (r = 0.34; *p* = 0.001), with Barthel index score (r = 0.27; *p* = 0.008) and with MUST questionnaire score (r = −0.22; *p* = 0.018)
Gencbas et al. (2018) [14]	NANDA-I Impaired urinary elimination (00016) NIC Urinary elimination management (0590) Urinary incontinenence care (0610) Urinary habit training (0600) Urinary bladder training (0570) Help with self-care: urination/defecation (1804) Environmental management (6480) Pelvic floor exercises (0560) Teaching: prescribed medication (5616) Urinary retention care (0620) NOC Urinary continence (0502) Urinary elimination (0503) Tissue Integrity: skin and mucous membranes (1101) Self-care: use of the toilet (0310) Response to medication (2301)	In the IG, NIC had the effect of improving all NOC scores following the intervention	
		NIC Urinary bladder training (0570) (*n* = 7)	NOC Urinary continence (Pre) M: 2.93; SD: 3.72 (Post) M: 4.41; SD: 0.24 (Difference) M: 1.48
			NOC Urinary elimination (Pre) M: 3.04; SD: 0.41 (Post) M: 4.49; SD: 0.22 (Difference) M: 1.45
		NIC Urinary elimination management (0590) (*n* = 32)	NOC Self-care: use of the toilet: Pre (M: 3.01; SD: 1.09); Post (M: 4.08; SD: 1.41); Difference M: 1.07
			NOC Urinary continence: Pre (M: 3.24; SD: 0.44); Post (M: 4.44; SD: 0.37); Difference M: 1.2
			NOC Urinary elimination: Pre (M: 3.23; SD: 0.46); Post (M: 4.59; SD: 0.22); Difference M: 1.36
		NIC Urinary habit training (0600) (*n* = 31)	NOC Urinary continence: Pre (M: 3.24; SD: 0.45); Post M: 4.45; SD: 0.37); Difference M: 1.21
			NOC Urinary elimination: Pre (M: 3.22; SD: 0.46); Post (M: 4.58; SD: 0.22); Difference M: 1.36
		NIC Help with self-care: urination/defecation (1804) (*n* = 29)	NOC Self-care: use of the toilet: Pre (M: 3.32; SD: 0.49); Post (M: 4.50; SD: 0.49); Difference M: 1.18
			NOC Urinary continence: Pre (M: 3.20; SD: 0.44); Post (M: 4.43; SD: 0.37); Difference M: 1.23
		NIC Environmental management (6480) (*n* = 29)	NOC Self-care: use of the toilet: Pre (M: 3.32; SD: 0.49); Post (M: 4.50; SD: 0.49); Difference M: 1.18
			NOC Urinary continence: Pre (M: 3.20; SD: 0.44); Post (M: 4.43; SD: 0.37); Difference M: 1.23
			NOC Urinary elimination: Pre (M: 3.17; SD: 0.44); Post (M: 4.57; SD: 0.22); Difference M: 1.4
		NIC Pelvic floor exercises (0560) (*n* = 32)	NOC Urinary continence: Pre (M: 3.24; SD: 0.44); Post (M: 4.44; SD: 0.37); Difference M: 1.2
			NOC Urinary elimination: Pre (M: 3.23; SD: 0.46); Post (M: 4.59; SD: 0.22); Difference M: 1.36
		NIC Urinary incontinence care (0610) (*n* = 32)	NOC Urinary continence: Pre (M: 3.24; SD: 0.44); Post (M: 4.44; SD: 0.37); Difference M: 1.2
			NOC Urinary elimination: Pre (M: 3.23; SD: 0.46); Post (M: 4.59; SD: 0.22); Difference M: 1.36
			NOC Tissue integrity: skin and mucous membranes: Pre (M: 4.10; SD: 0.75); Post M: 4.93; SD: 0.06); Difference M: 0.83
		NIC Teaching: prescribed medication (5616) (*n* = 7)	NOC Response to medication: Pre (M: 4.19; SD: 0.81); Post (M: 4.89; SD: 0.90); Difference M: 0.70
		NIC Urinary retention care (0620) (*n* = 7)	NOC Urinary continence: Pre (M: 3.12; SD: 0.26); Post (M: 4.48; SD: 0.21); Difference M: 1.36
Sampaio et al. (2018) [15]	NANDA-I Anxiety (00146) NIC Anxiety reduction (5820) Improvement of coping (5230) Relaxation therapy (6040) Assessment (5240) Help with anger control (4640) Intervention in case of crisis (6160) Reduction in stress due to relocation (5350) NOC Level of anxiety (1211) Self-control of anxiety (1402)	Favorable effect of the NIC on the NOC score	NOC Level of anxiety (d = 1.11) NOC Self-control of anxiety (d = 1.65)
		Being part of the IG predicts level of anxiety	22.8% (R2 adjusted: 0.228) Posttest (F (1.58) = 18.40); *p* < 0.001
		Moderate positive association between the variable “group” and the NOC Level of anxiety total score (1211) (posttest)	Β = 0.49
		Being part of the IG predicts self-control of anxiety	40% (R2 adjusted = 0.400) Posttest (F (1.58) = 40.27; *p* < 0.001)
		Moderate positive association between the variable “group” and total score in NOC Self-control of anxiety (posttest)	Β = 0.64
		NOC Level of anxiety (1211): mean differences by groups pre and post intervention	CG vs. IG: pretest CG (M: 34.58; SD: 8.91); pretest IG (M: 34.34; SD: 9.41); *p* = 0.92 CG (*n* = 31): pretest (M: 34.58; SD: 8.91); posttest (M: 45.71; SD: 12.36); *p* = 0.001 IG (*n* = 29): pretest (M: 34.34; SD: 9.41); posttest (M: 58.59; SD: 10.77); *p* = 0.001 CG vs. IG: posttest CG (M: 45.71; SD: 12.36); posttest IG (M: 58.59; SD: 10.77); *p* = 0.001
		NOC Self-control of anxiety (1402): Mean differences by groups pre and post intervention	CG vs. IG: pretest CG (M: 26.55; SD: 5.99); pretest IG (M: 27.1; SD: 4.81); *p* = 0.70 CG (*n* = 31) pretest (M: 26.55; SD: 5.99); posttest (M: 25.65; SD: 5.77); *p* = 0.55 IG (*n* = 29) pretest (M: 27.1; SD: 4.81); posttest (M: 34.21; SD: 4.57); *p* = 0.001 CG vs. IG: posttest CG (M: 25.6; SD: 5.77); posttest IG (M: 34.21; SD: 4.57); *p* = 0.001
Laguna-Parras et al. (2013) [28]	NANDA-I Sleep pattern disorder (00198) NIC Sleep improvement (1850) NOC Sleep (0004)	Oviedo Sleep Questionnaire: Satisfaction with sleep	(Admission) M: 3.27; SD: 1.51 (Discharge) M: 5.19; SD: 1.3 (Difference) M: 1.921; SD: 1.781; (CI: 1.71–2.12) *p* < 0.0001
		Oviedo Sleep Questionnaire: Insomnia	(Admission) M: 23.52; SD: 9.05 (Discharge) M: 15.93; SD: 8.25 (Difference) M: −7.59; SD: 10.95 (CI: 6.31–8.86) *p* < 0.0001
		Oviedo Sleep Questionnaire: Hypersomnia	(Admission) M: 5.97; SD: 3.76 (Discharge) M: 4.49; SD: 2.55 (Difference) M: −1.479; SD: 3.82 (CI: 1.03–1.92) *p* < 0.0001
		NOC Sleep (0004)	(Admission) M: 1.36; SD: 0.56 (Discharge) M: 3.84; SD: 0.68 (Difference) M: 2.48; SD: 0.84 (CI: 2.38–2.58) *p* < 0.0001

NNN: NANDA-NIC-NOC; NIC: Nursing Interventions Classification; NOC: Nursing Outcome Classification; RR: Relative risk; NNT: Number needed to treat; CG: Control group; IG: Intervention group; M: Mean; SD: Standard deviation; T1, T2, T3, T4, T5: Timepoint 1, 2, 3, 4, 5; PR: Prevalence ratio; CI: Confidence interval; BMI: Body mass index.

## 4. Discussion

Brazil is the country with the greatest number of publications, showing a marked tendency to explore aspects related to the clinical applicability of NNN, while Spain ranked second with a distinct emphasis on the growing interest in the study of nursing terminologies in our environment. The increase in the use and effectiveness of nursing SLSs in clinical practice is accompanied by improvements in the diagnostic reasoning capacities of the nurses [25].

Regarding the quality of evidence in these studies, the use of traditional systems such as the proposal by JBI to establish the LE has been refined with the application of GRADE methodology such that it is possible to adjust the focus and quality of the initial evidence rating granted according to the design of these studies’ methodologies, readjusting the factors or domains that confer the final certainty of the evidence to reduce it (assessing the risk of bias, inconsistency, indirectness, inaccuracy and publication bias) or increase it (assessing the magnitude of the effect, response gradient and absence of residual confounding) with greater certainty [29,30]. According to GRADE methodology, an RCT starts from high LE (1c according to JBI), thus the Corcoles et al. [12] study maintains high certainty of this LE; however, this certainty of LE decreases in the RCT carried out by Guerra et al. [13], Vázquez-Sánchez et al. [27], Sampaio et al. [15] and Müller-Staub et al. [23] to low certainty due methodological limitations (risk of bias, indirectness and imprecision). These aspects make it necessary to improve the rigour of the design of these studies. In contrast, cohort studies, which start from a lower LE according to JBI (3c: cohort with control group; 3d: case control; and 3e: observational without control group) and a low certainty of evidence according to GRADE, increased to a high certainty of LE in the studies of Pascoal et al. [17], Pascoal et al. [19,21] and Reis and Jesus [20]. The presence of these methodological weaknesses in the designs of the included studies, combined with the fact that each of these studies addressed different NNN concepts, has contributed to the heterogeneity of the findings, making it not possible to carry out comparative analyses of the measures of effect.

As background to this research, a study conducted by Müller-Staub et al. [30] assessed, among other aspects, the accuracy of the Standardized Nursing Terminology, in addition to the coherence between diagnoses, interventions and people’s health results. The authors identified deficits in the diagnostic process as well as in the notification of signs, symptoms and aetiologies, arguing for the need to implement training measures that ensure accuracy in nurses’ diagnostic reasoning [31,32]. To complement these criteria, the present study adds the importance of linking nurses’ critical thinking to the use of clinical indicators based on the best scientific evidence available from the results of rigorous research.

With respect to diagnostic etiological association, all the assumptions studied indicated that exposure to the aetiologies (related factors and risk factors) are diagnostic indicators for the presence of the health problems identified. The nursing diagnosis of risk of delayed surgical recovery (00246) includes people aged over 80 years in the NANDA-I classification, although the study only reported results that indicated an absence of statistical significance in this population with extreme ages. In contrast, the remaining aetiologies presented showed semantic variations.

Concerning the diagnosis of dysfunctional ventilatory response to weaning (00034), most of the statistically significant RFs reported by Silva et al. [16] were not included.

Regarding the analysis of diagnostic accuracy through the study of DCs, all studies were conducted about respiratory diseases by the same authors, and the presence of the DCs identified in these health problems were key indicators in each of the nursing diagnoses. The diagnosis of impaired gas exchange (00030) showed that abnormal skin colour and hypoxemia indicate the presence of this health issue with greater statistical accuracy. The 2021–2023 NANDA-I edition [5] includes these major DCs, which have high predictive value. A considerable number of minor or secondary DCs, with less predictive value for clinical judgement, have also been included. As such, it would be beneficial to add diagnostic accuracy criteria that distinguish between major and minor DCs to NANDA-I. The diagnosis of ineffective airway clearance (00031) showed that the only DCs not included in the 2021–2023 NANDA-I edition [5] correlate significant with open eyes, albeit with an excessively wide CI. On the other hand, the diagnosis of ineffective respiratory pattern (00032) showed effectiveness for diagnostic accuracy in all DCs, including others that were not observed in the study, suggesting that it would be valuable in future research to assess the rest of the DCs included in NANDA-I.

The assessment of changes in people’s health using NOC terminology has shown that planned interventions in clinical settings with specific diseases and certain risk situations using SLSs provide tools for the correct planning of nursing care. However, the literature supporting the use of these NOC indicators with validated tools providing objective data is limited; only the study conducted by Laguna-Parras et al. [28] has evaluated the NOC sleep (0004) for the diagnosis of sleep pattern disorder (00198) using the Oviedo Sleep Questionnaire.

In the effectiveness analysis for the resolution of specific health issues, certain modifications or the elimination of some diagnoses in the latest published edition of NANDA-I were notable [5]. Thus, functional urinary incontinence (00020) was replaced by another diagnosis called disability associated urinary incontinence (00297). Similarly, in the 2021–2023 NANDA-I edition, the diagnosis risk of falls (00155) was removed from the classification and replaced by new diagnoses which distinguish between the population of adults and children, with the diagnoses risk of falls in adults (00303) and risk of falls in children (00306). Likewise, for the diagnosis dysfunctional ventilatory response to weaning (00034), the 2021–2023 NANDA-I edition included diagnoses called dysfunctional ventilatory response to adult weaning, which differs from the previous definition by specifying that it refers to individuals over 18 who required mechanical ventilation for at least 24 h.

Only two studies, Cárdenas-Valladolid et al. [22] and Müller-Staub et al. [23], addressed general aspects of the use of NNN in the NP showing results supporting the use of the NP with NNN to improve clinical indicators in diabetes control with planned follow-up and increasing reasoning ability after a training program, respectively. In recent years, there has been growing interest among nurses in studying the clinical application of NNN. On the other hand, recent studies have shown more rigorous methodological designs, including cohort studies with adequate follow-up and randomized interventions with control groups that estimate the risk of bias. However, it is still essential to diversify international contexts and sample sizes in the populations studied with the aim of increasing effect measures in the population. Separately, it is vital that the results of these studies are transferred more quickly to the subsequent published NNN editions in order to improve nurses’ clinical impact.

In relation to diagnostic aetiology, this review has only assessed the association of RFs with clinical decision-making to identify nursing diagnoses; further studies should analyse the effects on the diagnostic accuracy of these aetiologies. In this sense, in the nursing field, a gold standard for diagnostic accuracy still needs to be developed. Moreover, this research has not addressed the existence of possible differences in relation to nurses’ gender and the use of SLSs. It would be interesting to develop future lines of research to explore differences between men and women in the application of the NP using NNN.

The limitations of the current research are due to the heterogeneity of the studies included in the SR, addressing distinct clinical situations corresponding to various health issues and NNN labels independently, which prevents comparison of results and the accumulated meta-analysis of their effect measures. Taking this into account, future research should examine larger sample sizes and the effect of longer follow-up periods in the populations studied.

## 5. Conclusions

It must be concluded that the scientific literature using NNN is very extensive but that there is still a deficit regarding the amount and quality of evidence and the degree of certainty concerning the effectiveness of the NP using these terminologies. At present, the use of NNN shows the clinical impact of nurses in health systems using SLSs; however, it is not yet possible to conclude that the use of NNN improves the effectiveness of the NP, besides in some rather specific clinical settings in which it has been assessed. The association between aetiologies and health problems identified by nurses is statistically significant in the few nursing diagnoses reviewed, but clinical decision-making must be studied in further nursing diagnoses. NANDA-I should update the diagnostic indicators in some diagnostic labels according to the evidence retrieved from the scientific literature. In addition, it is essential to approach diagnostic accuracy and the health results in people using NNN terminologies from the clinical perspective.

Most of studies reviewed have been based on the use of NNN in disease situations, so there is a need to develop more studies analysing the use of these terminologies in health promotion, community health and public health contexts. Similarly, it is important to implement the findings of new studies that assess the use of these terminologies with respect to improvements in the efficacy of nursing interventions and the satisfaction of the population with the NP. Finally, further methodologically rigorous studies are needed in a large number of clinical settings.

## Figures and Tables

**Figure 1 healthcare-11-02449-f001:**
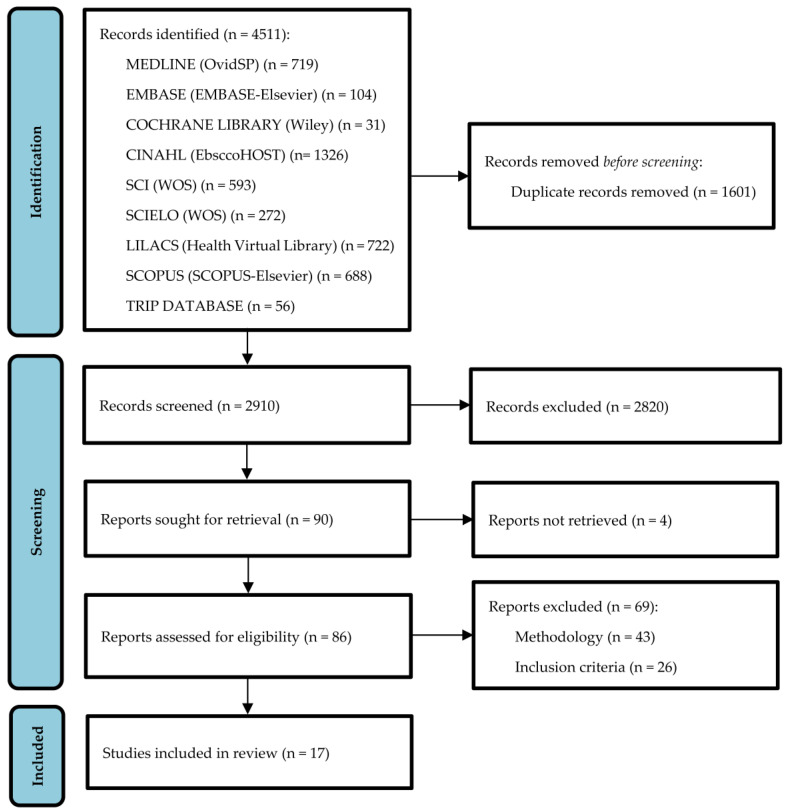
Flow chart.

## Data Availability

No new data were created.

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
