# Peer review of "Effectiveness of a Standardized Nursing Process Using NANDA International, Nursing Interventions Classification and Nursing Outcome Classification Terminologies: A Systematic Review"

_healthcare, 2023, doi:10.3390/healthcare11172449_

Round 1

Reviewer 1 Report

Dear authors, thank you for this important review manuscript. 

In general I congratulate the team for the very well structured and scientific soundness and methodological processes presented in the manuscript. 

I would like to suggest 3 litle improvements that your manuscript deserves regarding its quality: 

1) In the abstract consider replace the sentence: "The nursing process in clinical practice can be assessed using standardized language systems." for "The nursing clinical decision-making, regarding diagnoses, interventions and outcomes, can be assessed using standardized language systems."

2) Introduce in the abstract a sentence explaining that NANDA-I, NIC and NOC are examples of taxonomies used worldwide in nursing clinical IT records. 

3) I notice that the major diagnosis, interventions and outcomes described are in the context of disease. Maybe it would be nice that in the results and conclusions, a mention to the need of more studies regarding health promotion, community health and public health contexts should be made. 

3) It would be good to have in the conclusions an answer to your research question! 

Just a few more attention to these litle details and your manuscript will be very good

Reviewer 2 Report

These are all the comments I can incorporate. The article is very correct The topic being addressed has a specific interest: assessing the opportunity offered by the NANDA-NIC-NOC taxonomy in the design of Nursing care plans. It is a study that uses a rigorous methodology that provides new information for Nursing. It would improve if the gender perspective were incorporated in the methodological design and in the analysis of the results. The conclusions are consistent with the stated objectives. Bibliographical references are current and appropriate.

I congratulate the authors of the article for the choice of topic and the methodological rigor with which they have approached it. I only have one suggestion for improvement for future research: it would be very interesting to approach your research topic with a gender perspective. Because, have the differences between men and women in the effectiveness of a standardized nursing process been explored using NANDA? It's a very good line of research.

Reviewer 3 Report

It is advisable to include in the inclusion criteria, articles with a publication date of 5 or 10 years prior to this one.

Reviewer 4 Report

Abstract

There should be a reference to the platform used for each database, specifically for CINAHL and MEDLINE. For example, place MEDLINE (PubMed). This allows us an understanding of the search strings used.

Background

The background lacks complexity and detail. It does not lead to the review question. For example, the idea present in lines 56-59 needs to be more detailed. This will help readers understand why this review is needed.

The review question is not accurate. Although it is possible to understand what the authors want to develop, the use of nursing diagnosis and interventions is misleading. Consider introducing a specific outcome, for example, nursing diagnosis accuracy. This is extremely important because the type of review used may vary on your goal. For example, if the authors seek accuracy, the review should be a diagnostic test accuracy systematic review, whilst if the goal is to assess effectiveness, then a systematic review of effectiveness is the approach to select.

There is no clear mention in the text that a search in the literature was performed to understand if a similar review is in progress or has been developed.

Methods

PRISMA is not a methodology for developing Systematic reviews but rather a method to report them. Can you clarify the type of approach that has been used? (ex: Cochrane, JBI)

An appendix with all the search strings should be included. As it is, we cannot understand if the search was well conducted, hindering the quality of the article. Lines 85-93 are not enough to assess the quality of the search strategy.

Inclusion criteria should clearly state the mnemonic selected and indicate more precisely what the authors seek.

Exclusion criteria appear to be redundant.

Line 121 - Can you explicit in more detail how the pilot was performed? For example, how many studies were assessed during this stage?

Line 126 - Do you mean "Data extraction"?

Line 132 - Change to "Data synthesis"

Results

The authors must explain why the reports were not retrieved. There must be evidence that the authors approached the authors or used other strategies.

It is impossible to understand the accuracy of the nursing diagnosis as no reference test is used. This connects with the recommendation given at the end of the background, which hinders the overall assessment of the accuracy and reliability of the diagnoses.

Discussion

There is no summary of findings through GRADE. 

With the exception of the limitations, there is no indication of the heterogeneity of the studies.

The discussion seems to present the results of individual studies narratively but does not appear to provide a comprehensive synthesis. The quality or strength of the evidence evaluated across studies is also unclear.

Overall, the work presented is no more than a narrative review, where statistical data is extracted. There appears to be a lot of confusion about the method, leading to a discussion that does not meet the requirements of a systematic review of effectiveness.

Line 89 "those MESH most in line" is not well-built.

Round 2

Reviewer 3 Report

ok

Author Response

Dear reviewer, thank you very much for reviewing the manuscript. 

Reviewer 4 Report

First, thank you for your changes, as it has considerably improved the manuscript.

However, by mixing multiple types of reviews, the manuscript loses quality. In this case, less is better. Splitting the article into two reviews would be more beneficial and clarify many of its issues.

Another major issue has to do with the fact that the authors are trying to assess diagnostic accuracy through related factors. This does not meet the criteria of a systematic review of diagnostic accuracy. As stated in the previous review, a gold standard is required to provide an analysis of the accuracy and this is not present throughout the text. The type of review that the authors appear to be analysing is more in line with a systematic review of etiology and risk. According to JBI's description of such a review "Systematic reviews of etiology and risk factors assess the relationship (association) between certain factors (whether genetic or environmental for example) and the development of a disease or condition or other health outcome. Systematic reviews underpin evidence-based healthcare", it is possible to understand that the authors are trying to analyse how the presence of certain factors (e.g. pain) associates to the presence of a condition/nursing diagnosis (e.g. Risk of delayed surgical recovery).
